# Associations between Triglyceride-Glucose Index and Micro- and Macro-Angiopathies in Type 2 Diabetes Mellitus

**DOI:** 10.3390/nu12020328

**Published:** 2020-01-26

**Authors:** Hsuan Chiu, Hui-Ju Tsai, Jiun-Chi Huang, Pei-Yu Wu, Wei-Hao Hsu, Mei-Yueh Lee, Szu-Chia Chen

**Affiliations:** 1School of Medicine, College of Medicine, Kaohsiung Medical University, Kaohsiung 80708, Taiwan; mickey990055@gmail.com; 2Department of Family Medicine, Kaohsiung Municipal Ta-Tung Hospital, Kaohsiung Medical University Hospital, Kaohsiung Medical University, Kaohsiung 80708, Taiwan; bankin_0920@yahoo.com.tw; 3Graduate Institute of Clinical Medicine, College of Medicine, Kaohsiung Medical University, Kaohsiung 80708, Taiwan; 4Research Center for Environmental Medicine, Kaohsiung Medical University, Kaohsiung 80708, Taiwan; scarchenone@yahoo.com.tw; 5Division of Nephrology, Department of Internal Medicine, Kaohsiung Medical University Hospital, Kaohsiung Medical University, Kaohsiung 80708, Taiwan; karajan77@gmail.com (J.-C.H.); wpuw17@gmail.com (P.-Y.W.); 6Faculty of Medicine, College of Medicine, Kaohsiung Medical University, Kaohsiung 80708, Taiwan; 7Department of Internal Medicine, Kaohsiung Municipal Siaogang Hospital, Kaohsiung Medical University, Kaohsiung 80708, Taiwan; my345677@yahoo.com.tw; 8Division of Endocrinology and Metabolism, Department of Internal Medicine, Kaohsiung Medical University Hospital, Kaohsiung Medical University, Kaohsiung 80708, Taiwan

**Keywords:** triglyceride-glucose index, microangiopathy, macroangiopathy, type 2 diabetes mellitus

## Abstract

The aim of this study was to investigate the associations between the triglyceride-glucose (TyG) index, a marker of insulin resistance, and microangiopathies and macroangiopathies in patients with type 2 diabetes mellitus (DM). Our study enrolled 1990 type 2 diabetic patients from local hospitals in Taiwan during the period of 2002–2004. Multivariate logistic regression analysis was used to identify the factors related to microangiopathies, macroangiopathiess and TyG index. The patients were stratified by TyG index quartile (Q1–Q4). Adjusted odds ratios (aORs) of albuminuria for Q3 versus Q1 and Q4 versus Q1 were 1.424 (95% confidence interval [CI], 1.062–1.910, *p* = 0.018) and 1.948 (95% CI, 1.437–2.642, *p* < 0.001), respectively. The aOR of cerebrovascular disease (CVA) was 2.264 for Q4 versus Q1 (95% CI, 1.243–4.122, *p* = 0.008), but there were no significant associations with diabetic retinopathy (DR), coronary artery disease (CAD) or peripheral artery occlusive disease (PAOD). Significant stepwise increases were found in CVA (*p* = 0.0013) and albuminuria ≥30 mg/g (*p* < 0.001) corresponding to TyG index quartiles, but not DR (*p* = 0.573), CAD (*p* = 0.880) or PAOD (*p* = 0.223). There was a significant association between a high TyG index and a high risk of microalbuminuria and CVA in the study cohort. Further studies to investigate the use of medications in patients with DM to prevent micro- and macro-angiopathies are thus warranted.

## 1. Introduction

The increasing global prevalence of type 2 diabetes mellitus (DM) places an ever-increasing burden on healthcare systems due to the disease and its complications [1]. Most cases of DM-associated morbidity and mortality are caused by macro- and micro-vascular complications, including peripheral artery occlusive disease (PAOD), coronary artery disease (CAD), diabetic retinopathy (DR), cerebrovascular disease (CVA), diabetic neuropathy, and diabetic nephropathy [2]. Moreover, vascular complications are associated with acquired blindness, advanced atherosclerosis, end-stage renal disease and various neuropathies in patients with diabetes [3]. Poor glycemic control, hypertension, insulin resistance (IR), duration of DM and dyslipidemia are well-known risk factors for both micro- and macro-angiopathies [4]. Therefore, it is crucial to identify new approaches to tackle or prevent the development of vascular complications among patients with type 2 DM as early as possible.

IR is thought to play an important role in promoting atherosclerosis, and to be a major risk factor for the development of cardiovascular disease (CVD) [1,5,6]. The homeostasis model assessment of IR (HOMA-IR) is a validated and widely used method to quantify IR [7]. However, the use of HOMA-IR in clinical practice is limited by the requirement to measure insulin level. Recently, the triglyceride-glucose (TyG) index was proposed as a simple and reliable surrogate marker of IR, and it has been reported to be closely correlated with the HOMA-IR index [8,9,10]. Previous clinical data have shown associations of the TyG index to the atherosclerotic cardiovascular diseases, including carotid atherosclerosis [11], coronary artery calcification [12], arterial stiffness [13,14], and CVD [15,16,17,18]. Additionally, our previous study showed that a higher TyG index was associated with an increased risk of cardiovascular events among patients with type 2 DM [19]. Furthermore, Zhao et al. [20] reported associations between micro- and macro-vascular damage and the TyG index in an elderly community-dwelling Chinese population. They found significant associations between a high TyG index and higher risks of nephric microvascular damage and arterial stiffness. Nonetheless, no relevant clinical investigation has focused on the impact of the TyG index on micro- and macro-vascular diseases in patients with DM.

The aim of the present study was to investigate the associations between the TyG index and microangiopathies (microalbuminuria and DR) and macroangiopathies (CAD, CVA and PAOD) in patients with type 2 DM in southern Taiwan.

## 2. Materials and Methods

### 2.1. Study Patients

This was a retrospective cross-sectional study. The design of this study was described in detail in a previous study [21]. In brief, patients with type 2 DM were enrolled from the diabetes outpatient clinics of two hospitals located in the south of Taiwan between April 2002 and November 2004. Patients were excluded if they: (1) were undergoing dialysis or had an estimated glomerular filtration rate (eGFR) <15 mL/min/1.73 m2; (2) had type 1 DM (defined as acute hyperglycemic symptoms and heavy ketonuria [≥3], diabetic ketoacidosis, or receiving insulin for 1 year after type 1 DM had been diagnosed); or (3) had received a renal transplantation (Figure 1). After applying the exclusion criteria, 1990 patients (856 males and 1134 females) were included in this study, with a mean age of 64.0 ± 11.3 years. The Institutional Review Board of Kaohsiung Medical University Hospital approved this study, and written informed consent was obtained from all of the patients. In addition, all of the clinical investigations were conducted in accordance with the principles expressed in the ***Declaration of Helsinki***.

### 2.2. Collection of Demographic, Medical, and Laboratory Data

The medical and demographic data included waist circumference, sex, age, and comorbidities, were collected from medical records and by interviewing the patients. Body mass index (BMI) was calculated as the weight in kilograms divided by the height in meters squared. Pulse pressure was defined as the difference between systolic and diastolic blood pressure. All of the study population was already receiving treatment for diabetes upon enrollment. Blood samples were collected following overnight fasting, and biomarkers including triglycerides, fasting glucose, HbA1c, creatinine and total cholesterol were measured on an autoanalyzer (COBAS Integra 400, Roche Diagnostics, Risch-Rotkreuz, Switzerland). The TyG index was calculated as ln fasting triglycerides mg/dL × fasting glucose mg/dL/2 [9]. The compensated Jaffé (kinetic alkaline picrate) method was used to measure the levels of serum creatinine on the Roche/Integra 400 Analyzer (Roche Diagnostics) using a calibrator that could be traced in isotope-dilution mass spectrometry [22]. The Modification of Diet in Renal Disease study equation [23] was used to calculate eGFR. The use of all medications during the study period, including statins, angiotensin II receptor blockers (ARBs), angiotensin converting enzyme inhibitors (ACEIs), and fibrates, was recorded from medical records.

### 2.3. Definition of Albuminuria

Microalbuminuria was defined as a urine albumin/creatinine ratio (UACR) ≥ 30 mg/g. An auto-analyzer (COBAS Integra 400 plus; Roche Diagnostics, Risch-Rotkreuz, Switzerland) was used to measure urine creatinine and albumin levels in spot urine samples.

### 2.4. Evaluation of DR

DR was diagnosed according to ophthalmic examinations including fundoscopy that were interpreted and approved by an ophthalmologist specialist and defined as being either proliferative or nonproliferative [24].

### 2.5. Definitions of CAD and CVA

CAD was defined according to statements made by the patients and hospital medical records of old myocardial infarction, ischemic electrocardiographic changes, angina, or having received coronary bypass surgery or angioplasty. CVA was also defined according to statements made by the patients and hospital medical records of cerebrovascular events including bleeding and cerebral infarction.

### 2.6. Assessment of Ankle–Brachial Index (ABI) and Definition of PAOD

The detailed methods of measuring the ABI have been described in previous studies [21,25,26]. ABI was measured once in each patient using an ABI-form device (VP1000; Omron Colin Co. Ltd., Komaki, Japan) and calculated as systolic blood pressure in the ankle/systolic blood pressure in the arm. The smallest systolic blood pressure in the ankle was used for the analysis [27,28,29]. In brief, after the patients had been placed in the supine position and allowed to rest for 10 min, monitoring and occlusion cuffs were attached firmly around the upper and lower extremities. An ABI of <0.9 in either leg was used to define PAOD.

### 2.7. Statistical Analysis

The studied patients were stratified into four groups according to TyG index quartile. The data are expressed as percentages for categorical variables or mean ± SD for continuous variables. One-way analysis of variance (ANOVA) and post hoc Bonferroni correction were used for among-group comparisons. We used multivariate forward logistic regression analysis to investigate factors associated with macro- and micro-angiopathies. A *p* value < 0.05 was considered to be statistically significant. All statistical analyses were performed using SPSS version 19.0 for Windows (IBM Corp, Armonk, NY, USA).

## 3. Results

This study enrolled 1990 participants stratified into four groups by TyG index quartile, with 497, 499, 495 and 499 patients in each, respectively. The clinical characteristics of these groups are shown and compared in Table 1. The cutoff values of the TyG index quartiles were <8.7, ≥8.7, <9.1, ≥9.1, <9.5, and ≥9.5, respectively. The mean values ± standard deviation of TyG index in the four groups were 8.3 ± 0.3, 8.9 ± 0.1, 9.3 ± 0.1, and 10.0 ± 0.5, respectively. The patients in quartile 4 had significantly higher systolic and diastolic blood pressures, higher BMI, higher fasting glucose, higher HbA1c, higher triglyceride level, higher total cholesterol level, higher prevalence of UACR > 30 mg/g, and higher rates of statin and fibrate use than those in quartile 1. We used trend analysis to investigate relationships between the quartiles of TyG index and micro- and macro-angiopathies. Significant trends in stepwise increases in albuminuria ≥ 30 mg/g (*p* for trend < 0.001) and CVA (*p* for trend = 0.013) corresponding to the quartiles of the TyG index were noted, but not DR (*p* for trend = 0.573), CAD (*p* for trend = 0.880), or PAOD (*p* for trend = 0.223) (Figure 2).

Table 2 shows the associations between the TyG index and other common risk factors and the prevalence of microangiopathies in our cohort. After adjusting for age, sex, pulse pressure, waist circumference, HbA1c, total cholesterol, BMI ≥ 25 kg/m^2^, eGFR and statin or fibrate use in multivariate analysis, the adjusted odds ratios (ORs) of albuminuria for quartile 2 vs. quartile 1, quartile 3 vs. quartile 1, and quartile 4 vs. quartile 1 were 1.215 (95% confidence interval [CI], 0.905–1.631, *p* = 0.196), 1.424 (95% CI, 1.062–1.910, *p* = 0.018), and 1.948 (95% CI, 1.437–2.642, *p* < 0.001), respectively. Moreover, there were independent associations between albuminuria ≥ 30 mg/g and low eGFR (95% CI, 0.856–0.952, *p* < 0.001), BMI ≥ 25 kg/m^2^ (95% CI, 1.115–1.683, *p* = 0.003), high HbA1c (95% CI, 1.099–1.249, *p* < 0.001), and wide pulse pressure (95% CI, 1.096–1.251, *p* < 0.001). In addition, wide pulse pressure, low cholesterol, low eGFR and high HbA1c were significantly associated with DR, but not the TyG index.

Table 3 shows the association between the TyG index and other common risk factors and the prevalence of macroangiopathies in our cohort. CVA was independently associated with male patients and old age. In the multivariate forward logistic regression analysis, the adjusted ORs of CVA for quartile 2 vs. quartile 1, quartile 3 vs. quartile 1, and quartile 4 vs. quartile 1 were 1.127 (95% CI, 0.585–2.171, *p* = 0.720), 1.618 (95% CI, 0.862–3.038, *p* = 0.134), and 2.264 (95% CI, 1.243–4.122, *p* = 0.008), respectively. There were also independent associations between CAD and BMI ≥ 25 kg/m^2^, old age, low eGFR and low cholesterol. In addition, ABI < 0.9 was also independently associated with low eGFR and old age. We used goodness-of-fit tests to decide whether the model was correctly specified. The global *p*-values estimated by Hosmer and Lemeshow goodness-of-fit tests in these models were below 0.05, which indicated that these models were convincing (Table 2 and Table 3).

## 4. Discussion

The TyG index has the advantage of being applicable in clinical practice since both triglyceride and glucose levels are routinely measured and the cost of these measurements is low. Several studies have reported the predictive ability of the TyG index in both patients with metabolic disorders and the general population [30,31,32,33]. However, no relevant clinical investigation has focused on whether the TyG index has a predictive role in identifying patients with type 2 DM at high risk of developing future micro- and macro-vascular diseases. In this study, we investigated associations between the TyG index and micro- and macro-angiopathies in patients with type 2 DM, and found that a higher TyG index was independently associated with microalbuminuria and CVA, but not DR, CAD or PAOD.

In our study, the first important finding is that a high TyG index is correlated with microalbuminuria. Diabetic nephropathy is a major manifestation of diabetic microangiopathies and is also the leading cause of end-stage renal disease worldwide [3]. In the early stage, diabetic nephropathy presents as microalbuminuria. Previous studies have reported an association between IR and an increased risk of developing progressive diabetic nephropathy among patients with diabetes [34,35]. The Northern Shanghai Study [20] also found that a higher TyG index was associated with a higher risk of microalbuminuria and chronic kidney disease in elderly Chinese community-dwelling individuals.

Different mechanisms may underlie the association between IR and microalbuminuria. IR has been shown to lead to hyperinsulinemia as a means to counteract the reduction in insulin action. Hyperinsulinemia has been reported to induce renal vasodilatation, elevate renal vascular permeability, raise glomerular hydrostatic pressure, enhance renal sodium reabsorption and increase glomerular hyperfiltration [36,37,38]. Nephron loss can be caused by an increase in filtration load, resulting in glomerular hypertension and sclerosis, and subsequently renal dysfunction [39]. In summary, the cascade of reactions caused by IR lead to a change in renal endothelial function and hemodynamics. This may explain the underlying mechanisms for the significant correlation between the TyG index and the development of microalbuminuria observed in this study.

DR affects the retinal vasculature and can lead to vision loss. The risk factors for DR include age, sex, glycemic control, type and duration of DM, hypertension, BMI, serum lipid levels, nephropathy, and smoking [40,41]. Previous studies have reported a positive association between IR and DR [42,43]. However, we did not identify a significant association between DR and the TyG index in this study. The major lipoprotein abnormalities in IR include a change in the size of low-density lipoprotein (LDL) particles, lower levels of high-density lipoprotein (HDL), and hypertriglyceridemia [44]. The relationship between DR and plasma lipid levels was inconsistent in previous studies. In the Multiethnic Study of Atherosclerosis [45], the authors did not identify associations between serum lipid levels (HDL, LDL and total cholesterol) and DR. Lyons et al. [46] reported a significant association between retinopathy and triglycerides, whereas Stratton et al. [47] did not find such as association. The retina is protected against potential damage caused by plasma lipoproteins by the blood–retinal barrier. Apolipoprotein B can damage capillaries in the retina, thereby leading to extravasation, and the severity of DR in patients with DM has been associated with the level of apolipoprotein B [48]. Taken together, these findings suggest that the dysregulation of intraretinal lipid transport may play a more important role than levels of plasma lipids in the pathogenesis of DR [49]. The TyG index is calculated using the plasma triglyceride level, and this may explain the weak association between triglycerides and DR in this study.

Another important finding of this study is the correlation between a high TyG index and CVA. Vascular complications, and particularly atherosclerosis, are major causes of morbidity and mortality in patients with DM. We found significant trends of stepwise increases in CVA corresponding to the quartile of the TyG index. Our previous study investigated the role of the TyG index in predicting cardiovascular events [19]. Several studies have reported a significant relationship between the TyG index and carotid artery atherosclerosis [11,50]. Nonetheless, previous research on the association between the incidence of stroke and the TyG index has reported inconsistent results. In a recent study on the Vascular-Metabolic CUN cohort [18], a higher TyG index resulted in a higher risk for incident ischemic stroke, whereas Zheng et al. did not find an association between the TyG index and CVA events [33]. Taken together, the results of the present study could be explained by increased IR and carotid atherosclerosis [6,51]. Insulin resistance was related to endothelial dysfunction, alterations in hemostatic process, and inflammation [51]. These pathogenic changes could increase the risk of CVD [51].

Few studies have reported associations between a high TyG index and the presence of coronary artery complications in type 2 DM and healthy adults [12,15,52]. In this study, we did not find a statistically significant association between the TyG index and CAD in the patients with type 2 DM. In addition, a recent study of 7129 Korean subjects also found no associations between IR parameters and a higher risk of CAD in patients with diabetes [53]. Importantly, they emphasized that the TyG index was only associated with an increased risk of CAD in non-diabetic patients. This finding could be explained by the use of medications. A meta-analysis showed that statin therapy had significant benefit on major vascular events in diabetic patients [54]. The Fenofibrate Intervention and Event Lowering in Diabetes (FIELD) study indicated that fenofibrate did not significantly decrease the risk of the primary outcome of major coronary events [55]. Moreover, fenofibrate could reduce the risk of total cardiovascular events and microvascular-related complications in patients with type 2 DM [56]. In the current study, the patients in quartile 4 of the TyG index had significantly higher rates of statin and fibrate use. After correlating statin and fibrate use, the significant relationship between the TyG index and CAD was not found.

In this study, we did not find a significant association between PAOD and the TyG index. Although many studies have established the association between type 2 DM and the risk of PAOD [56,57], few studies have investigated the association between IR and PAOD. Vonbank et al. [58] reported that IR, measured by HOMA-IR, was not directly linked to sonographically proven peripheral arterial disease in 214 Caucasian patients with intermittent claudication. Conversely, the National Health and Nutrition Examination Survey [59] and the Cardiovascular Health Study [60] demonstrated positive interrelationships between IR and PAOD in a general population as measured by HOMA-IR and ABI, respectively. Since increasing age and the duration of DM are also important risk factors for the development of PAOD [56], patient selection may have contributed to the difference between their findings and ours. This hypothesis was supported by the recent Northern Shanghai Study [20], which showed that higher quartiles of the TyG index were associated with an increased risk of lower extremity atherosclerosis (defined as ABI < 0.9) in univariate logistic regression. Nonetheless, after adjusting for traditional risk factors, the correlation between the TyG index and PAOD was not statistically significant.

There are several limitations to this study. First, we did not screen the patients according to the duration of disease, and this may have influenced the associations between DM and macro- and micro-angiopathies. DM duration, especially in type 2 diabetes, is a very unreliable subjective statement from patients, and this can lead to bias in studies of diabetes. Second, because our participants were all of Chinese ethnicity, our findings may not be generalizable to other ethnicities. Third, we did not include the HOMA-IR index in our analysis, which is the gold standard method to measure IR. However, as mentioned, the close relationship between the TyG index and HOMA-IR has already been well established. Moreover, the TyG index is easier to calculate and less expensive in routine clinical practice. Furthermore, the definitions, medications and treatment algorithms of DM were different at that time, which may need to be considered. Finally, due to the design of this study, we could not make definitive conclusions with regard to causal relationships and long-term clinical outcomes. However, we believe that our findings may provide evidence of the importance of the TyG index in patients with DM at risk of both microangiopathies and macroangiopathies. Further prospective studies are warranted to investigate associations between the TyG index and the development and progression of micro- and macro-angiopathies in patients with DM. 

## 5. Conclusions

In this study of patients with type 2 DM, we found that a high TyG index was associated with an increased risk of microalbuminuria and CVA, but not DR, CAD or PAOD. We suggest that educational interventions should be provided for patients with a high TyG index with regard to lifestyle changes, along with more intensive clinical interventions if necessary. 

## Figures and Tables

**Figure 1 nutrients-12-00328-f001:**
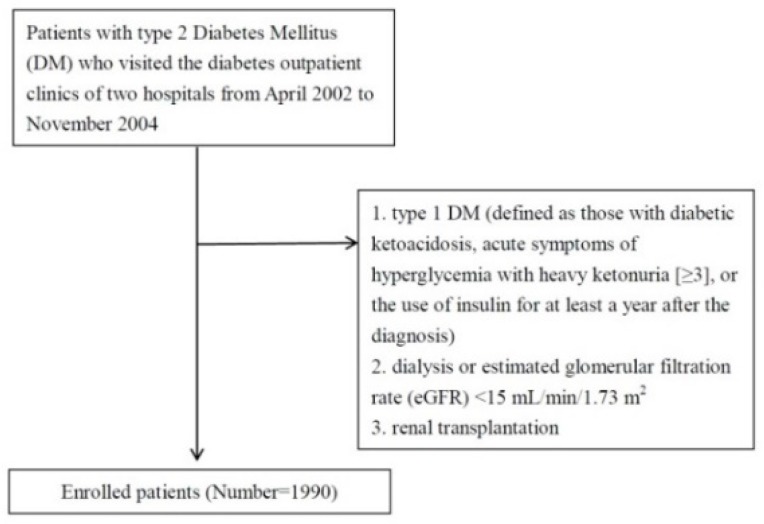
Flowchart.

**Figure 2 nutrients-12-00328-f002:**
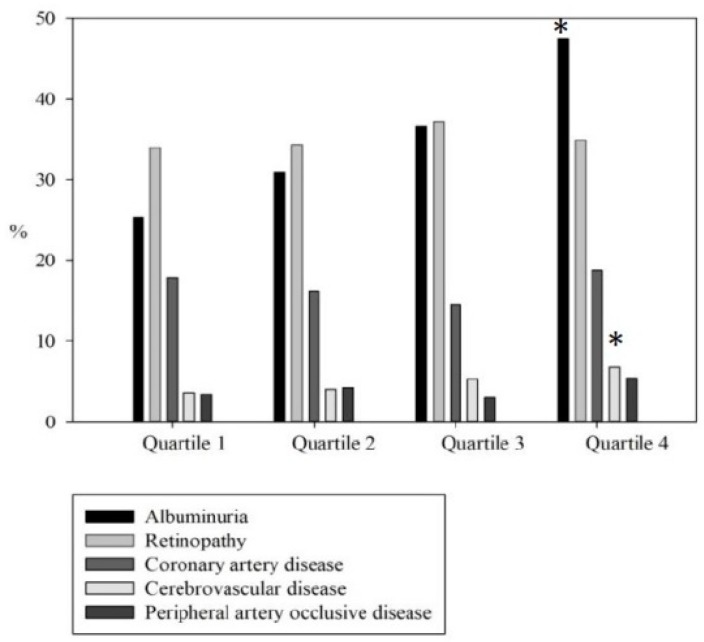
There were significant trends for stepwise increases in albuminuria ≥ 30 mg/g (*p* for trend < 0.001) and cerebrovascular disease (*p* for trend = 0.013) corresponding to quartiles of TyG index, but not diabetic retinopathy (*p* for trend = 0.573), coronary artery disease (*p* for trend = 0.880), and peripheral artery occlusive disease (*p* for trend = 0.223).

**Table 1 nutrients-12-00328-t001:** Comparison of Baseline Characteristics According to TyG index Quartile.

Characteristics	Quartile 1	Quartile 2	Quartile 3	Quartile 4
	(<8.7)	(≥8.7, <9.1)	(≥9.1, <9.5)	(≥9.5)
	*n* = 497	*n* = 499	*n* = 495	*n* = 499
TyG index	8.3 ± 0.3	8.9 ± 0.1 ^a^	9.3 ± 0.1 ^ab^	10.0 ± 0.5 ^abc^
Age (years)	64.4 ± 12.1	64.7 ± 10.5	64.1 ± 11.4	63.1 ± 11.2
Male (%)	45.9	46.1	38.6	41.5
Systolic blood pressure (mmHg)	132.9 ± 19.6	134.3 ± 18.2	135.6 ± 17.5	136.8 ± 19.5 ^a^
Diastolic blood pressure (mmHg)	75.2 ± 10.6	78.1 ± 11.1 ^a^	78.2 ± 10.8 ^a^	79.8 ± 12.3 ^a^
Pulse pressure (mmHg)	57.7 ± 16.4	56.2 ± 14.7	57.4 ± 14.6	57.0 ± 15.3
Body mass index (kg/m^2^)	24.8 ± 3.4	26.0 ± 3.6 ^a^	26.2 ± 3.5 ^a^	26.4 ± 3.6 ^a^
Waist circumference (cm)	89.4 ± 47.8	90.1 ± 9.6	92.3 ± 40.8	90.8 ± 9.2
Laboratory parameters				
Fasting glucose (mg/dL)	115.6 ± 30.3	134.6 ± 31.6 ^a^	154.4 ± 41.9 ^ab^	189.5 ± 62.6 ^abc^
HbA1c (%)	7.0 ± 1.2	7.3 ± 1.3 ^a^	7.7 ± 1.5 ^ab^	8.6 ± 2.0 ^abc^
TG (mg/dL)	77.0 ± 23.0	114.2 ± 27.4 ^a^	149.7 ± 38.7 ^ab^	278.7 ± 217.8 ^abc^
Total cholesterol (mg/dL)	172.0 ± 30.5	177.7 ± 29.8 ^a^	188.0 ± 34.7 ^ab^	204.9 ± 48.3 ^abc^
eGFR (mL/min/1.73 m^2^)	70.7 ± 19.0	69.3 ± 19.3	67.0 ± 19.8 ^a^	67.6 ± 20.4
Albuminuria ≥ 30 mg/g (%)	25.4	30.9	36.6 ^a^	47.5 ^abc^
DR (%)	34.0	34.3	37.2	34.9
CAD (%)	17.9	16.2	14.5	18.8
CVA (%)	3.6	4.0	5.3	6.8
ABI < 0.9 (%)	3.4	4.2	3.0	5.4
Medications				
ACEI and/or ARB (%)	70.2	75.6	73.8	74.8
Statin use (%)	46.6	58. 5 ^a^	63.3 ^a^	71.2 ^ab^
Fibrate use (%)	4.7	8.2	14.6 ^ab^	39.0 ^abc^

Abbreviations: ABI = ankle-brachial index; ACEI = angiotensin-converting enzyme inhibitor; ARB = angiotensin II receptor blocker; CAD = coronary artery disease; CVA = cerebrovascular disease; DR = diabetic retinopathy; eGFR = estimated glomerular filtration rate; HbA1c = glycated hemoglobin A1c; TG = triglyceride; TyG = triglyceride-glucose. The TyG index was calculated as ln (fasting triglyceride [mg/dL] × fasting glucose [mg/dL]/2); ^a^
*p* < 0.05 vs. quartile 1, ^b^
*p* < 0.05 vs. quartile 2, ^c^
*p* < 0.05 vs. quartile 3.

**Table 2 nutrients-12-00328-t002:** Determinants of Microangiopathies Using Forward Binary Logistic Regression Analysis ^a^.

Microangiopathies	Multivariate ^a^	
	OR (95% CI)	*p*
Albuminuria ≥ 30 mg/g ^b^		
TyG index		
Quartile 1	Reference	
Quartile 2	1.215 (0.905–1.631)	0.196
Quartile 3	1.424 (1.062–1.910)	0.018
Quartile 4	1.948 (1.437–2.642)	<0.001
Pulse pressure (per 10 mm Hg)	1.171(1.096–1.251)	<0.001
Body mass index ≥ 25 kg/m^2^	1.370 (1.115–1.683)	0.003
HbA1c (per 1%)	1.172 (1.099–1.249)	<0.001
eGFR (per 10 mL/min/1.73 m^2^)	0.903(0.856–0.952)	<0.001
DR ^c^		
Pulse pressure (per 10 mm Hg)	1.139(1.068–1.215)	<0.001
HbA1c (per 1%)	1.106 (1.040–1.176)	0.001
Cholesterol (per 1 mg/dL)	0.997 (0.995–1.000)	0.044
eGFR (per 10 mL/min/1.73 m^2^)	0.882(0.838–0.929)	<0.001

Abbreviations: CI = confidence interval; DR = diabetic retinopathy; eGFR = estimated glomerular filtration rate; HbA1c = glycated hemoglobin A1c; OR = odds ratio; TG =triglyceride; ^a^ Covariates in the multivariate model included quartiles of TyG index, age, sex, pulse pressure, body mass index ≥ 25 kg/m^2^, waist circumference, HbA1c, total cholesterol, eGFR, and statin or fibrate use. ^b^ Global *p* value estimated by Hosmer and Lemeshow Goodness-of-Fit Test = 0.810. ^c^ Global *p* value estimated by Hosmer and Lemeshow Goodness-of-Fit Test = 0.556.

**Table 3 nutrients-12-00328-t003:** Determinants of Macroangiopathies Using Forwad Binary Logistic Regression Analysisa.

Macroangiopathies	Multivariate ^a^	
	OR (95% CI)	*p*
CAD ^b^		
Age (per 1 year)	1.062 (1.046–1.077)	<0.001
Body mass index ≥ 25 kg/m^2^	1.380 (1.062–1.793)	0.016
Cholesterol (per 1 mg/dL)	0.996 (0.992–0.999)	0.020
eGFR (per 10 mL/min/1.73 m^2^)	0.917 (0.851–0.989)	0.024
CVA ^c^		
TyG index		
Quartile 1	Reference	
Quartile 2	1.127 (0.585–2.171)	0.720
Quartile 3	1.618 (0.862–3.038)	0.134
Quartile 4	2.264 (1.243–4.122)	0.008
Age (per 1 year)	1.059 (1.036–1.082)	<0.001
Male (vs female)	1.686 (1.101–2.581)	0.016
ABI < 0.9 ^d^		
Age (per 1 year)	1.048 (1.019–1.077)	0.001
eGFR (per 10 mL/min/1.73 m^2^)	0.702 (0.607–0.811)	<0.001

Abbreviations: ABI = ankle-brachial index; CAD = coronary artery disease; CI = confidence interval; CVA = cerebrovascular disease; eGFR = estimated glomerular filtration rate; OR = odds ratio; ^a^ Covariates in the multivariate model included quartiles of TyG index, age, sex, pulse pressure, body mass index ≥ 25 kg/m^2^, waist circumference, HbA1c, total cholesterol, eGFR, and statin or fibrate use. ^b^ Global *p* value estimated by Hosmer and Lemeshow Goodness-of-Fit Test = 0.558. ^c^ Global *p* value estimated by Hosmer and Lemeshow Goodness-of-Fit Test = 0.297. ^d^ Global *p* value estimated by Hosmer and Lemeshow Goodness-of-Fit Test = 0.488.

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
