# Peer review of "Associations between Triglyceride-Glucose Index and Micro- and Macro-Angiopathies in Type 2 Diabetes Mellitus"

_nutrients, 2020, doi:10.3390/nu12020328_

Round 1
Reviewer 1 Report
The report from Chiu and colleagues evaluates a possible association between Triglyceride Glucose (TyG) index and micro- and macro-angiopathies in type 2 diabetes patients. The study population is well-defined, the purpose of the study is clinically significant, and the manuscript is well written.
I have some minor suggestions for improvement of the manuscript:
-How is “pulse pressure” defined? Please state in the methods section
-Line 27 (Abstract section): the authors enrolled 1,990 patients and not 1,981. Please, replace 1,981 with 1,990.
-Line 88: please correct the formula by adding brackets to the product and by replacing log with ln for better clarity
-Table 1: when defining the quartile ranges, does 9.1 belong to the 2nd or 3rd quartile? Please correct.
-Lines 130-133 and Figure 1: how are the p-values calculated? Please specify in the methods section and in the figure legend. Moreover, the chart type should be stated as a first sentence in the legend. Which data are shown by the graph?
-Line 146 and line 159: The term “risk factor” could be misleading. I know you are referring to the common risk factors for microalbuminuria and DR, but your study actually evaluates the association between the reported variable and the prevalence of the aforementioned complications. Therefore you could update the sentences as “Table X shows the association of TyG index and other common risk factors and the prevalence of micro/macroangiopathies in our cohort”.
-Table 2 and 3: add the Beta-values for each variable
-Table 2 and 3: please add the overall model fit and global p-value (to the legend or the main text) for each logistic regression analysis.
-Table 2: the authors should consider recalculating the ORs for pulse pressure and eGFR by considering larger increments, such as 10 mmHg or 10 mL/min.
-Discussion, line 180. Replace the word “role” with “association”. You did not evaluate the (predictive) role of TyG. The association was evaluated instead.
-In my opinion, the discussion section is too long. The authors should consider shortening of the comparisons with previous literature.
Author Response
Response to Reviewer 1 Comments
The report from Chiu and colleagues evaluates a possible association between Triglyceride Glucose (TyG) index and micro- and macro-angiopathies in type 2 diabetes patients. The study population is well-defined, the purpose of the study is clinically significant, and the manuscript is well written.
I have some minor suggestions for improvement of the manuscript:
Point 1: How is “pulse pressure” defined? Please state in the methods section
Response 1: Pulse pressure is the difference between systolic and diastolic blood pressure. We add this description in the “Materials and Methods”, Line 101
Point 2: Line 27 (Abstract section): the authors enrolled 1,990 patients and not 1,981. Please, replace 1,981 with 1,990.
Response 2: We revised it. (Line 27)
Point 3: Line 88: please correct the formula by adding brackets to the product and by replacing log with ln for better clarity
Response 3: We revised it. (Line 105)
Point 4: Table 1: when defining the quartile ranges, does 9.1 belong to the 2nd or 3rd quartile? Please correct.
Response 4: We revised it as ≥ 8.7, < 9.1; ≥ 9.1, <9.5 (Table 1 and Line 144-145)
Point 5: Lines 130-133 and Figure 1: how are the p-values calculated? Please specify in the methods section and in the figure legend. Moreover, the chart type should be stated as a first sentence in the legend. Which data are shown by the graph?
Response 5: Thanks for your comment. We revised it. Please see “Result” line 149-153 and figure 2.
We used trend analysis to investigate the relationship between the quartiles of TyG index and micro- and macro-angiopathies. There were significant trends for stepwise increases in albuminuria ≥ 30 mg/g (p for trend < 0.001) and CVA (p for trend= 0.013) corresponding to the quartiles of TyG index, but not DR (p for trend= 0.573), CAD (p for trend= 0.880), or PAOD (p for trend= 0.223) (Figure 2).
Point 6: Line 146 and line 159: The term “risk factor” could be misleading. I know you are referring to the common risk factors for microalbuminuria and DR, but your study actually evaluates the association between the reported variable and the prevalence of the aforementioned complications. Therefore you could update the sentences as “Table X shows the association of TyG index and other common risk factors and the prevalence of micro/macroangiopathies in our cohort”.
Response 6: We have revised it. Please see “result”, Line 181 & 198
Table 2 shows the association of TyG index and other common risk factors and the prevalence of microangiopathies in our cohort. (Line181);
Table 3 shows the association of TyG index and other common risk factors and the prevalence of macroangiopathies in our cohort. (Line198).
Point 7: Table 2 and 3: add the Beta-values for each variable
Response 7: We stated odds ratio (OR) in multivariate logistic regression model.
Point 8: Table 2 and 3: please add the overall model fit and global p-value (to the legend or the main text) for each logistic regression analysis.
Response 8: Thanks for your suggestion. We have added the overall model fit and global p-value in main text (line 190) and legend (line 205 & 213)
We used goodness-of-fit tests to decide whether your model is correctly specified. The global p-values estimated by Hosmer and Lemeshow goodness-of-fit test in these models were below 0.05, which indicated these models were convince (Table 2 and Table 3).
Table 2
bGlobal p value estimated by Hosmer and Lemeshow Goodness-of-Fit Test = 0.915
cGlobal p value estimated by Hosmer and Lemeshow Goodness-of-Fit Test = 0.418
Table 3
bGlobal p value estimated by Hosmer and Lemeshow Goodness-of-Fit Test = 0.549
cGlobal p value estimated by Hosmer and Lemeshow Goodness-of-Fit Test = 0.482
dGlobal p value estimated by Hosmer and Lemeshow Goodness-of-Fit Test = 0.279
Point 9: Table 2: the authors should consider recalculating the ORs for pulse pressure and eGFR by considering larger increments, such as 10 mmHg or 10 mL/min.
Response 9: Thanks for your suggestion. We revised Table 2 and Table 3.
Table 2. Determinants of Microangiopathies Using Forward Binary Logistic Regression Analysisa.
|
Microangiopathies |
Multivariatea |
|
|
|
OR (95% CI) |
P |
|
Albuminuria ≥30 mg/gb |
|
|
|
TyG index |
|
|
|
Quartile 1 |
Reference |
|
|
Quartile 2 |
1.227 (0.914-1.648) |
0.173 |
|
Quartile 3 |
1.446 (1.079-1.938) |
0.014 |
|
Quartile 4 |
1.971 (1.454-2.672) |
< 0.001 |
|
Pulse pressure (per 10 mm Hg) |
1.174(1.098-1.247) |
< 0.001 |
|
Body mass index ≥ 25 kg/m2 |
1.363 (1.110-1.674) |
0.003 |
|
HbA1c (per 1%) |
1.117 (1.098-1.247) |
< 0.001 |
|
eGFR (per 10 mL/min/1.73 m2) |
0.905(0.858-0.954) |
< 0.001 |
|
DRc |
|
|
|
Pulse pressure (per 10 mm Hg) |
1.139(1.068-1.215) |
< 0.001 |
|
HbA1c (per 1%) |
1.108 (1.042-1.177) |
0.001 |
|
Cholesterol (per 1 mg/dL) |
0.997 (0.995-1.000) |
0.046 |
|
eGFR (per 10 mL/min/1.73 m2) |
0.883(0.838-0.930) |
< 0.001 |
Abbreviations: CI = confidence interval; DR = diabetic retinopathy; eGFR = estimated glomerular filtration rate; HbA1c = glycated hemoglobin A1c; OR = odds ratio; TG =triglyceride;
aCovariates in the multivariate model included quartiles of TyG index, age, sex, pulse pressure, body mass index ≥ 25 kg/m2, waist circumference, HbA1c, total cholesterol, and eGFR
bGlobal p value estimated by Hosmer and Lemeshow Goodness-of-Fit Test = 0.915
cGlobal p value estimated by Hosmer and Lemeshow Goodness-of-Fit Test = 0.418
Table 3. Determinants of Macroangiopathies Using Forwad Binary Logistic Regression Analysisa.
|
Macroangiopathies |
Multivariatea |
|
|
|
OR (95% CI) |
P |
|
CADb |
|
|
|
Age (per 1 year) |
1.061 (1.046-1.077) |
< 0.001 |
|
Body mass index ≥ 25 kg/m2 |
1.392 (1.072-1.809) |
0.013 |
|
Cholesterol (per 1 mg/dL) |
0.996 (0.9992-0.999) |
0.021 |
|
eGFR (per 10 mL/min/1.73 m2) |
0.917 (0.851-0.988) |
0.023 |
|
CVAc |
|
|
|
TyG index |
|
|
|
Quartile 1 |
Reference |
|
|
Quartile 2 |
1.134 (0.589-2.183) |
0.708 |
|
Quartile 3 |
1.614 (0.860-3.029) |
0.136 |
|
Quartile 4 |
2.271 (1.248-4.134) |
0.007 |
|
Age (per 1 year) |
1.059 (1.036-1.082) |
< 0.001 |
|
Male (vs female) |
1.684 (1.100-2.578) |
0.016 |
|
ABI < 0.9d |
|
|
|
Age (per 1 year) |
1.046 (1.018-1.076) |
0.001 |
|
eGFR (per 10 mL/min/1.73 m2) |
0.683 (0.592-0.789 |
< 0.001 |
Abbreviations: ABI = ankle-brachial index; CAD = coronary artery disease; CI = confidence interval; CVA = cerebrovascular disease; eGFR = estimated glomerular filtration rate; OR = odds ratio; aCovariates in the multivariate model included quartiles of TyG index, age, sex, pulse pressure, body mass index ≥ 25 kg/m2, waist circumference, HbA1c, total cholesterol, and eGFR.
bGlobal p value estimated by Hosmer and Lemeshow Goodness-of-Fit Test = 0.549
cGlobal p value estimated by Hosmer and Lemeshow Goodness-of-Fit Test = 0.482
dGlobal p value estimated by Hosmer and Lemeshow Goodness-of-Fit Test = 0.279
Point 10: Discussion, line 180. Replace the word “role” with “association”. You did not evaluate the (predictive) role of TyG. The association was evaluated instead.
Response 10: We revised it. We evaluated the association of the TyG index in microangiopathies and macroangiopathies in patients with type 2 DM (line 222).
Point 11: In my opinion, the discussion section is too long. The authors should consider shortening of the comparisons with previous literature.
Response 11: Thanks for your suggestion. We have revised it. Please see “Discussion”
Reviewer 2 Report
The work presents a structure according to what is required by the magazine, but requires several changes. The first thing you need is an update of the biobliography to more up-to-date appointments of the last five years. In the material and methods section it would be necessary to add which organization has approved the work prior to its realization as well as the corresponding ethics or bioethics committee. As for the results, they are quite clear and understandable, but susceptible to improvement.
Author Response
Response to Reviewer 2 Comments
The work presents a structure according to what is required by the magazine, but requires several changes.
Point 1: The first thing you need is an update of the biobliography to more up-to-date appointments of the last five years.
Response 1: Thanks for your suggestion. We have revised our reference.
Point 2: In the material and methods section it would be necessary to add which organization has approved the work prior to its realization as well as the corresponding ethics or bioethics committee.
Response 2: Thanks for your comment. We stated it in Material and methods line 82-85: The protocol was approved by the Institutional Review Board of Kaohsiung Medical University Hospital. Written informed consent was obtained from the patients, and all clinical investigations were conducted according to the principles expressed in the Declaration of Helsinki and line 317, Funding: This research was funded by KAOHSIUNG MUNICIPAL SIAOGANG HOSPITAL, KMHK-A-107-026, KMHK-A-107-002 and KAOHSIUNG MEDICAL UNIVERSITY HOSPITAL, SA10802, KMU-DK108004.
Point 3: As for the results, they are quite clear and understandable, but susceptible to improvement.
Response 3: We have revised it
Reviewer 3 Report
No comments for the authors.
Author Response
Response to Reviewer 3 Comments
Point 1: No comments for the authors
Response 1: Thanks for your comment. (in red)
Reviewer 4 Report
Thank you for your earnest work. Triglyceride-glucose index has become a new tool for predicting prognoses of different cohorts of patients. Here are my suggestions for the manuscript:
Major problems:
Please make a flowchart of the study design before data presentations.
The enrollment period was more than 15 years before. The definitions, medicines and treating algorithms of DM were obviously different at that time so the results might be outdated and inconvincible.
You have mentioned several times that the duration of DM was a major factor affecting the formation of angiopathies. It was a bias rather than a limitation. It would make all results unreliable if this variable was not adjusted.
CAD and CVA were major outcomes in the study. Please take the medicines such as antiplatelets or anticoagulants which might affect the outcomes of those diseases into correlation. In addition, you should explore the association between TGI and different types of stroke.
In line 146, you have adjusted multiple variables before analysis. Therefore the adjusted variables including PP, BMI, HbA1c, and eGFR should not be found in table 2. Table 3 has the same mistake. Please recheck the methods with the statisticians again.
Please supplement figures to emphasize your study. Pearson correlation test or Cox proportional regression model might be appropriate for your final results.
Minor problems:
Please describe the design of the study. Was it retrospective or prospective? Was it a cohort study or a cross-sectional study?
Please clarify the definitions of the outcomes. Was the diabetic retinopathy defined by ophthalmic examinations or just ICD codes? Were the diagnoses of CAD and CVA defined by admission histories or just ICD codes?
Please describe the collect timing of urine. Was it collected before, during or after DM treatments?
In line 111, patients with vascular access were excluded from performing VP1000. However, patients of eGFR less than 15 or ever receiving kidney transplantation were excluded already before ABI performance. There might be few patients with vascular access after the enrollment. Please show the real number of patients with access or omit the description.
Please describe the limitations of the study in detail.
Please recheck your description in line 60. It was not compatible with the 18th reference.
Please rephrase sentences in line 56-58 and line 173-174 to avoid duplication. In addition, a mistake of uppercase was found in line 166.
Author Response
Response to Reviewer 4 Comments
Major problems:
Point 1: Please make a flowchart of the study design before data presentations.
Response 1: Thanks for your suggestion. We add flowchart in Figure 1.
Point 2: The enrollment period was more than 15 years before. The definitions, medicines and treating algorithms of DM were obviously different at that time so the results might be outdated and inconvincible.
Response 2: Thanks for your comment.
Our objectives was not attempted to investigate the effects of new medication or treatment on diabetes mellitus. Therefore, the enrollment period might be not a major factor that can affect our outcome study in micro- and macroangiopathies. The definitions on micro- and macroangiopathies were not redefined for the past decades. However, the diagnostic criteria of DM included HbA1c ≥ 6.5 since 2010 ADA guideline, and more DM patients were found. The number of this study is sufficient to analyse, and our result may be significant in more participants.
We have added your suggestion in limitation, line 300-301
Besides, the definitions, medicines and treating algorithms of DM were different at that time, which may need to be considered.
Point 3: You have mentioned several times that the duration of DM was a major factor affecting the formation of angiopathies. It was a bias rather than a limitation. It would make all results unreliable if this variable was not adjusted.
Response 3: Thanks for your precise comment. We agreed with your comments on the DM duration was a bias in the study of diabetes. DM duration, especially in type 2 diabetes is a very unreliable subjective statement from the patient, which might mislead for us in studying this type of population. Even in the RCT studying in cardiovascular outcome, DM duration is only a reference in comparing the difference between two study groups in baseline demographic data, but rarely be an adjusting factor. We have stated it in limitation, line 292-295.
First, the patients were not screened according to disease duration, and this may have influenced the effects of DM on micro angiopathies and macroangiopathies. DM duration, especially in type 2 diabetes is a very unreliable subjective statement from the patient. DM duration was a bias in the study of diabetes.
Point 4: CAD and CVA were major outcomes in the study. Please take the medicines such as antiplatelets or anticoagulants which might affect the outcomes of those diseases into correlation. In addition, you should explore the association between TGI and different types of stroke.
Response 4: Thank you for kind comments, but the incidence of stroke in our study population was as low as to only in the range of 3.6-6.8%. The statistics were not able to reach significance if we split it to 2 types of stroke due to the little population. According to the guidelines in treatment of diabetes mellitus, antiplatelet nor antithrombitic agents were not recommended routinely use as preventive medication for stroke or CAD, therefore we were not able to put these agents as important as the evidenced drugs for prevention for CAD and Stroke like statin and fibrates.
Point 5: In line 146, you have adjusted multiple variables before analysis. Therefore the adjusted variables including PP, BMI, HbA1c, and eGFR should not be found in table 2.
Table 3 has the same mistake. Please recheck the methods with the statisticians again.
Response 5: We used multivariate forward logistic regression model to adjust quartiles of TyG index, age, sex, pulse pressure, body mass index ≥ 25 kg/m2, waist circumference, HbA1c, total cholesterol, and eGFR. The significant variables were shown in Table 2 & Table 3.
Point 6: Please supplement figures to emphasize your study. Pearson correlation test or Cox proportional regression model might be appropriate for your final results.
Response 6: Thanks for your suggestion. Figure 2 showed that there were significant trends for stepwise increases in albuminuria ≥ 30 mg/g (p for trend < 0.001) and cerebrovascular disease (p for trend = 0.013) corresponding to quartiles of TyG index, but not diabetic retinopathy (p for trend = 0.573), coronary artery disease (p for trend = 0.880), and peripheral artery occlusive disease (p for trend = 0.223).
We added the supplement figure 1 to show the correlation between TyG index and ACR, TyG index and ABI using Pearson correlation test.
Supplement figure 1. The correlation between TyG index and ACR, TyG index and ABI
Minor problems:
Point 1: Please describe the design of the study. Was it retrospective or prospective? Was it a cohort study or a cross-sectional study?
Response 1: This was retrospective cross-sectional study. We have added it in Methods (line 75).
Point 2: Please clarify the definitions of the outcomes. Was the diabetic retinopathy defined by ophthalmic examinations or just ICD codes? Were the diagnoses of CAD and CVA defined by admission histories or just ICD codes?
Response 2: We have revised it in Methods (line 118 & 122).
The diabetic retinopathy was defined by ophthalmic examinations done by fundoscopy, interpreted and approved by ophthalmologist specialist.
CAD was defined by the statement of the patients and medical records in our hospitals of old myocardial infarction, ischemic electrocardiographic changes, angina, or having received angioplasty or coronary bypass surgery. CVA was defined by the statement of the patients and medical records in our hospitals of cerebrovascular events including cerebral infarction and bleeding.
Point 3: Please describe the collect timing of urine. Was it collected before, during or after DM treatments?
Response 3: We have revised it. Please see Line 102.
Our study population was already in the course of treatment for diabetes upon enrollment.
Point 4: In line 111, patients with vascular access were excluded from performing VP1000. However, patients of eGFR less than 15 or ever receiving kidney transplantation were excluded already before ABI performance. There might be few patients with vascular access after the enrollment. Please show the real number of patients with access or omit the description.
Response 4: Thanks for your correction. We have excluded patients of eGFR less than 15 or ever receiving kidney transplantation. We revised the description (line 131: monitoring and occlusion cuffs were attached firmly around the upper and lower extremities)
Point 5: Please describe the limitations of the study in detail.
Response 5: We have revised the limitation
Point 6: Please recheck your description in line 60. It was not compatible with the 18th reference.
Response 6: We have revised it. Please see line 64
Zhao et al. [19] reported associations between micro- and macro-vascular damage and the TyG index in an elderly community-dwelling Chinese population.
Point 7: Please rephrase sentences in line 56-58 and line 173-174 to avoid duplication. In addition, a mistake of uppercase was found in line 166.
Response 7: We revised it.
Reviewer 5 Report
This is the interesting clinical study addressing the triglyceride-glucose index and micro- and macro-angiopathies in humans with type 2 diabetes.
The study is well performed, however, minor points should be improved.
Line 64 and 178
Authors indicate that the clinical study addressing the triglyceride-glucose index and micro- and macro-angiopathies in patients with type 2 diabetes is lacking. However, the same group of Authors (Jiun-Chi Huang, Pei-Yu Wu, Wei-Hao Wsu, Mey-Yueh Lee and Szu-Chia Chen) published the article concerning the potential link between triglyceride-glucose index and cardio-vascular events in humans with type 2 diabetes.
Su WY, Chen SC, Huang YT, Huang JC, Wu PY, Hsu WH, Lee MY: Comparison of the Effects of Fasting Glucose, Hemoglobin A1c, and Triglyceride-Glucose Index on Cardiovascular Events in Type 2 Diabetes Mellitus. Nutrients 2019 19;11(11).
However, this article is not included and discussed in the present paper.
Fig. 1. Statistical differences are not marked.
Table 2 and 3. The title of both tables is inadequate since parameters other than determinants of microangiopathies and macroangiopathies are also included.
Some minor corrections should be made.
For example, Line 124 coma instead of point.
Line 195 The sentence is not clear.
Author Response
Response to Reviewer 5 Comments
This is the interesting clinical study addressing the triglyceride-glucose index and micro- and macro-angiopathies in humans with type 2 diabetes.
The study is well performed, however, minor points should be improved.
Point 1: Line 64 and 178
Authors indicate that the clinical study addressing the triglyceride-glucose index and micro- and macro-angiopathies in patients with type 2 diabetes is lacking. However, the same group of Authors (Jiun-Chi Huang, Pei-Yu Wu, Wei-Hao Wsu, Mey-Yueh Lee and Szu-Chia Chen) published the article concerning the potential link between triglyceride-glucose index and cardio-vascular events in humans with type 2 diabetes.
Su WY, Chen SC, Huang YT, Huang JC, Wu PY, Hsu WH, Lee MY: Comparison of the Effects of Fasting Glucose, Hemoglobin A1c, and Triglyceride-Glucose Index on Cardiovascular Events in Type 2 Diabetes Mellitus. Nutrients 2019 19;11(11).
However, this article is not included and discussed in the present paper.
Response 1: We have revised it. Please see Introduction line 62 and Discussion line 265
Additionally, our previous study showed that a higher TyG index was associated with an increased risk of cardiovascular events among patients with type 2 DM [18].
Our previous study has investigated the role of the TyG index in predicting cardiovascular events [18].
Point 2: Fig. 1. Statistical differences are not marked.
Response 2: We have revised it
Point 3: Table 2 and 3. The title of both tables is inadequate since parameters other than determinants of microangiopathies and macroangiopathies are also included.
Response 1: We have revised it.
Point 4: For example, Line 124 coma instead of point.
Response 4: We have revised it
Point 5: Line 195 The sentence is not clear.
Response 5: We have revised it
Round 2
Reviewer 4 Report
I appreciate your prompt and detailed responses. Here are some additional comments for the revision:
In the statistical analysis section, why did you choose forward logistic regression analysis rather than the backward or bidirectional method to avoid suppressor effects? Please explain the reason.
You have mentioned that medicines including statins and fibrates were evident to prevent CAD and CVA in the diabetic cohort. The percentage of medicine usage was significantly different among quartiles. Why didn’t you correlate these items in the multivariate regression model? Additionally, you should cite the reference for the benefits of medicines.
Author Response
I appreciate your prompt and detailed responses. Here are some additional comments for the revision:
Point 1. In the statistical analysis section, why did you choose forward logistic regression analysis rather than the backward or bidirectional method to avoid suppressor effects? Please explain the reason.
Response 1: Thanks for your precise comment. The forward and backward logistic regression models were useful in clinical research. We performed backward logistic regression analysis in eTable 2 and eTable 3. These results in backward logistic regression model were the same as forward logistic regression model.
eTable 2. Determinants of Microangiopathies Using Backward Binary Logistic Regression Analysisa.
|
Microangiopathies |
Multivariatea |
|
|
|
OR (95% CI) |
P |
|
Albuminuria ≥30 mg/gb |
|
|
|
TyG index |
|
|
|
Quartile 1 |
Reference |
|
|
Quartile 2 |
1.215 (0.905-1.631) |
0.196 |
|
Quartile 3 |
1.424 (1.062-1.910) |
0.018 |
|
Quartile 4 |
1.948 (1.437-2.642) |
< 0.001 |
|
Pulse pressure (per 10 mm Hg) |
1.171(1.096-1.251) |
< 0.001 |
|
Body mass index ≥ 25 kg/m2 |
1.370 (1.115-1.683) |
0.003 |
|
HbA1c (per 1%) |
1.172 (1.099-1.249) |
< 0.001 |
|
eGFR (per 10 mL/min/1.73 m2) |
0.903(0.856-0.952) |
< 0.001 |
|
DRc |
|
|
|
Pulse pressure (per 10 mm Hg) |
1.139(1.068-1.215) |
< 0.001 |
|
HbA1c (per 1%) |
1.106 (1.040-1.176) |
0.001 |
|
Cholesterol (per 1 mg/dL) |
0.997 (0.995-1.000) |
0.044 |
|
eGFR (per 10 mL/min/1.73 m2) |
0.882(0.838-0.929) |
< 0.001 |
eTable 3. Determinants of Macroangiopathies Using Backward Binary Logistic Regression Analysisa.
|
Macroangiopathies |
Multivariatea |
|
|
|
OR (95% CI) |
P |
|
CADb |
|
|
|
Age (per 1 year) |
1.062 (1.046-1.077) |
< 0.001 |
|
Body mass index ≥ 25 kg/m2 |
1.380 (1.062-1.793) |
0.016 |
|
Cholesterol (per 1 mg/dL) |
0.996 (0.992-0.999) |
0.020 |
|
eGFR (per 10 mL/min/1.73 m2) |
0.917 (0.851-0.989) |
0.024 |
|
CVAc |
|
|
|
TyG index |
|
|
|
Quartile 1 |
Reference |
|
|
Quartile 2 |
1.127 (0.585-2.171) |
0.720 |
|
Quartile 3 |
1.618 (0.862-3.038) |
0.134 |
|
Quartile 4 |
2.264 (1.243-4.122) |
0.008 |
|
Age (per 1 year) |
1.059 (1.036-1.082) |
< 0.001 |
|
Male (vs female) |
1.686 (1.101-2.581) |
0.016 |
|
ABI < 0.9d |
|
|
|
Age (per 1 year) |
1.048 (1.019-1.077) |
0.001 |
|
eGFR (per 10 mL/min/1.73 m2) |
0.702 (0.607-0.811) |
< 0.001 |
|
Statin or Fibrate use |
2.170 (1.171-4.021) |
0.014 |
Point 2. You have mentioned that medicines including statins and fibrates were evident to prevent CAD and CVA in the diabetic cohort. The percentage of medicine usage was significantly different among quartiles. Why didn’t you correlate these items in the multivariate regression model? Additionally, you should cite the reference for the benefits of medicines.
Response 2: Thanks for your precise comment. We add “statin or fibrate use” in the multivariate regression model and revise Table 2 and Table 3 according to your suggestion. Our results were similar after adjusting “statin or fibrate use”. We also revised the manuscript in Line 279-285 and cited the reference as 54&55
A meta-analysis showed that statin therapy had significant benefit on major vascular events in diabetic patients [54]. The Fenofibrate Intervention and Event Lowering in Diabetes (FIELD) study indicated that fenofibrate did not significantly decrease the risk of the primary outcome of major coronary events [55]. Moreover, fenofibrate could reduce the risk of total cardiovascular events and microvascular-related complications in patients with type 2 DM [55]. In the current study, the patients in quartile 4 of the TyG index had significantly higher rates of statin and fibrate use. After correlating statin and fibrate use, the significant relationship between the TyG index and CAD was not found.
- Cholesterol Treatment Trialists' (CTT) Collaborators.; Kearney, P.M.; Blackwell, L.; Collins, R.; Keech, A.; Simes, J.; Peto, R.; Armitage, J.; Baigent, C. Efficacy of cholesterol-lowering therapy in 18,686 people with diabetes in 14 randomised trials of statins: a meta-analysis. Lancet. 2008, 371, 117-125.
- Keech, A.; Simes, R.J.; Barter, P.; Best, J.; Scott, R.; Taskinen, M.R.; Forder, P.; Pillai, A.; Davis, T.; Glasziou, P.; Drury, P.; Kesäniemi, Y.A.; Sullivan, D.; Hunt, D.; Colman, P., d'Emden, M.; Whiting, M.; Ehnholm, C.; Laakso, M.; FIELD study investigators. Effects of long-term fenofibrate therapy on cardiovascular events in 9795 people with type 2 diabetes mellitus (the FIELD study): randomised controlled trial. Lancet. 2005, 366, 1849-1861.
Table 2. Determinants of Microangiopathies Using Forward Binary Logistic Regression Analysisa.
|
Microangiopathies |
Multivariatea |
|
|
|
OR (95% CI) |
P |
|
Albuminuria ≥30 mg/gb |
|
|
|
TyG index |
|
|
|
Quartile 1 |
Reference |
|
|
Quartile 2 |
1.215 (0.905-1.631) |
0.196 |
|
Quartile 3 |
1.424 (1.062-1.910) |
0.018 |
|
Quartile 4 |
1.948 (1.437-2.642) |
< 0.001 |
|
Pulse pressure (per 10 mm Hg) |
1.171(1.096-1.251) |
< 0.001 |
|
Body mass index ≥ 25 kg/m2 |
1.370 (1.115-1.683) |
0.003 |
|
HbA1c (per 1%) |
1.172 (1.099-1.249) |
< 0.001 |
|
eGFR (per 10 mL/min/1.73 m2) |
0.903(0.856-0.952) |
< 0.001 |
|
DRc |
|
|
|
Pulse pressure (per 10 mm Hg) |
1.139(1.068-1.215) |
< 0.001 |
|
HbA1c (per 1%) |
1.106 (1.040-1.176) |
0.001 |
|
Cholesterol (per 1 mg/dL) |
0.997 (0.995-1.000) |
0.044 |
|
eGFR (per 10 mL/min/1.73 m2) |
0.882(0.838-0.929) |
< 0.001 |
Table 3. Determinants of Macroangiopathies Using Forwad Binary Logistic Regression Analysisa.
|
Macroangiopathies |
Multivariatea |
|
|
|
OR (95% CI) |
P |
|
CADb |
|
|
|
Age (per 1 year) |
1.062 (1.046-1.077) |
< 0.001 |
|
Body mass index ≥ 25 kg/m2 |
1.380 (1.062-1.793) |
0.016 |
|
Cholesterol (per 1 mg/dL) |
0.996 (0.992-0.999) |
0.020 |
|
eGFR (per 10 mL/min/1.73 m2) |
0.917 (0.851-0.989) |
0.024 |
|
CVAc |
|
|
|
TyG index |
|
|
|
Quartile 1 |
Reference |
|
|
Quartile 2 |
1.127 (0.585-2.171) |
0.720 |
|
Quartile 3 |
1.618 (0.862-3.038) |
0.134 |
|
Quartile 4 |
2.264 (1.243-4.122) |
0.008 |
|
Age (per 1 year) |
1.059 (1.036-1.082) |
< 0.001 |
|
Male (vs female) |
1.686 (1.101-2.581) |
0.016 |
|
ABI < 0.9d |
|
|
|
Age (per 1 year) |
1.048 (1.019-1.077) |
0.001 |
|
eGFR (per 10 mL/min/1.73 m2) |
0.702 (0.607-0.811) |
< 0.001 |